# Structural-Temporal Peculiarities of Dynamic Deformation of Layered Materials

**DOI:** 10.3390/ma15124271

**Published:** 2022-06-16

**Authors:** Nina Selyutina, Yuri Petrov

**Affiliations:** Mathematical and Mechanical Faculty, Saint Petersburg State University, Universitetskiy Prospect 28, Saint Petersburg 198504, Russia; y.v.petrov@spbu.ru

**Keywords:** strain rate, layered composites, deformation curve, relaxation plasticity model, fibre, stress drop

## Abstract

The temporal nature of static and dynamic deformation of fibre metal laminates is discussed here. The aim of the study is to verify the proposed innovate model using layered composites. The modified relaxation model is based on the earlier formulated plasticity relaxation model for homogeneous materials. The proposed relaxation model makes it possible to describe the deformation of the layered composites from elastic to irreversible deformation, finalised by the failure moment. The developed approach allows us to consider the effects of the transition from static to dynamic loading. This means that the model-calculated dynamic limiting characteristics of the metal and the strength of brittle materials will have a determining character, depending on the loading history. The verification of the model using a glass fibre reinforced aluminium composite, glass fibre reinforced titanium composite, carbon fibre reinforced aluminium composite, and Kevlar fibre reinforced aluminium composite with different thickness ratios between metal and polymer layers is given. It is shown that the theoretical deformation curves of the metal composites at the various strain rates, finalised by brittle fracture of the polymer layers or continued irreversible deformation of remaining unbroken metal layers with destroyed polymer (fibre/epoxy) layers, are predicted. Based on the same structural−temporal parameters for five (Ti/GFRP (0/90)/Ti/GFRP(90/0)/Ti) and three (Ti/GFRP(0/90/90/0)/Ti) layers glass fibre reinforced titanium composites and the polymer layers, one-stage and two-stage stress drops during the irreversible deformation of the composite under static and dynamic loading are simulated. The change of the multi-stage fracture of the composite from static to dynamic loading and the fracture characteristic times of the polymer (100 s and 15,400 s) and the metal (8.4 ms) are correlated. Continued plastic deformation of the composite after fracture of the polymer layers is related with different values of the characteristic relaxation times of the polymer (fibre/epoxy) and the metal layers.

## 1. Introduction

Two types of theories are used to numerically simulate the nonlinear initial deformation of the layered metal composite structures under static loads. The first type [1,2] considers the composite as a combination of two homogeneous materials, and through mixing of the deformation dependencies of brittle polymer layers and ductile metal layers, the complete deformation dependence of the composite is modelled. In the classical theory of laminates [3,4,5] (the second most used type), the deformation of the composite is determined through the plane stress state, and the deformation dependence is calculated according to Hooke’s law by creating a stiffness matrix of the composite. Both types of theories apply to polymer or metal matrix laminated composites, but are not used in modeling the static failure of concrete composites with different types of aggregates [6,7,8]. Using the finite element method without failure criteria, it is possible to simulate the initial deformation response of the composite [9]. The multi-stage process of composite fracture under static loads can be predicted using various nonlinear damage models [10,11,12]. Determining the contact condition between the layers, it is possible to simulate the deformation of the composite before failure using the finite element method [13,14].

The multi-stage process of the fracture of the multilayer metal composites leads to the dynamic deformation dependencies of the metal composites differing from the typical deformation dependencies of homogeneous materials [15,16]. The simultaneous deformation of plastic and brittle materials complicates the preparatory process of fracture, and new types of deformation diagrams arise under both static and dynamic loading, depending on the combination of materials in the composite [17,18,19,20]. The appearance of the time factor in the case of irreversible deformation of the composite under dynamic loads requires the expansion of typical elastic criteria for plastic materials [21,22,23], as well as the introduction of new characteristics that take into account the temporal or strain rate features of the deformation of materials.

To formulate the deformation model of the layered metal composites subjected to dynamic loads, the classical Johnson−Cook model for plastic deformation of metals, the Johnson−Cook damage model, and the delamination model are used [24,25]. The number of fitting parameters that do not have a physical meaning increases when considering the deformation process under dynamic loads. The study shows that numerical models cannot be used for various homogeneous materials under static and dynamic loads, as the parameters of these models depend on the loading history. In other words, for each load range, a different dynamic plasticity model for homogeneous materials must be selected.

The studies listed above show the impossibility of using some of the unified approaches for predicting the deformation of multilayer composites. When studying composite materials under dynamic loads, new difficulties are added related to the dependence of the material response on the applied loading rate and generally on the loading history. For homogeneous materials, there is also no unified approach for calculating the deformation of a material under a dynamic load, as there are many different constitutional models. The main difficulty in using dynamic models for homogeneous materials is the binding of one material to a model and the applicability of the model in a certain range of strain rates. Undoubtedly, the integral approach of Tuler and Butcher shows a good result [26], but there is no understanding regarding determining the stable parameters of the material responsible for the dynamic response of the material when using this model. Thus, the development of a unified dynamic model for the deformation of multilayer composites remains an urgent task today.

In this paper, the development of our idea, proposed earlier in [27,28], by applying the relaxation model of heterogeneous materials, is continued. The verification of the proposed model for composites (glass fibre reinforced aluminium composite, glass fibre reinforced titanium composite, carbon fibre reinforced aluminium composite, and Kevlar fibre reinforced aluminium composite) tested under dynamic and static conditions is given. The proposed model is based on the relaxation plasticity model for homogeneous materials [29,30,31] and the fracture criteria for homogeneous materials [32,33]. Using the proposed model, it is possible to calculate the elastic, inelastic, and irreversible stages of composite deformation until the composite is completely destroyed. The proposed relaxation model makes it possible to describe the deformation of the layered composites from elastic to irreversible deformation, finalised by the failure moment. The developed approach allows for considering the effects of the transition from static to dynamic loading. It means that the model-calculated dynamic limiting characteristics of the metal and the strength of brittle materials will have a determining character, depending on the loading history. The peculiarity of the model used is the use of invariant parameters to the deformation history, called the structural−temporal approach. When modelling the fracture process, the structural and temporal parameters of both the composite itself and its composite materials are taken into account. It is assumed that the proposed relaxation model for the heterogeneous materials takes into account the strain rate effects at different loading histories, as well as the relaxation model for homogeneous materials. The model predicts the deformation dependence, which is typical for brittle and ductile−brittle fractures of the composite materials with different structural and temporal parameters. So, it has been proven that the idea of using a united set of characteristic times of plasticity and fracture, which previously worked well for predicting ductile−brittle transitions [34,35] and dynamic phase transformations [33], also makes it possible to effectively predict important competitive phenomena in materials with a complex internal structure.

## 2. Fracture Criterion of Brittle Materials

To predict the strength of brittle materials in a wide range of strain rates, the structural−temporal approach [32,33] is considered, based on the idea of a material incubation time [35]. The corresponding fracture criteria proposed in [36,37,38,39,40] allowed for studying a large set of different phenomena of the dynamic brittle fracture process successfully. The set of material parameters in this approach is invariant to the history of loading, *Σ*(*t*). These parameters allow for comparing the dynamic characteristics of the material obtained from experiments with different shapes of the loading pulse [41]. The fracture condition in our particular case is determined by the incubation time criterion, which can be represented by the following relation:(1)$$\varphi \left(t,\Sigma \left(t\right),\sigma_{f},\tau_{f},\alpha_{f}\right)\leq 1,\;\;\;\varphi \left(\theta ,L\left(\theta \right),a_{*},a_{IT}^{1},a_{IT}^{2}\right)=\frac{1}{a_{IT}^{1}}\;\int_{\theta -a_{IT}^{1}}^{\theta }{\left(\frac{L\left(\theta_{1}\right)}{a_{*}}\right)}^{a_{IT}^{2}}d\theta_{1}$$
where *τ* is the incubation time of fracture associated with the dynamics of the micro-cracking relaxation processes preceding the macro-fracture event, *Σ*(*t*) is the time dependence of the local tensile stress in the specimen, and *σ_f_* is the static strength of the brittle fibres. The time of fracture *t_f_* is defined as the moment at which the equality sign in Equation (1) is attained. The parameter α characterizes the sensitivity of the material to the level (amplitude) of force field causing the fracture (or structural transformation). The set of material constants *τ* and α determines the fracture process on a given scale level.

## 3. Plasticity Relaxation Model for Homogeneous Materials

The relaxation model of plasticity [29,30,31] for the development of the structural−temporal approach for plasticity [34,42] is based on the material incubation time concept [42]. The use of the material incubation time to describe the temporal effects of plastic deformation considers shear stress relaxation to be a temporal process related to the development and motion of defects [29,30]. The relaxation itself can be realized by various physical mechanisms, depending on the particular material. In terms of the material incubation time, the relaxation mechanism is not explicitly described, but it is stated that it requires some characteristic time owing to the development and motion of microdefects.

The dimensionless relaxation function *γ*(*t*) (0 < *γ*(*t*) ≤ 1) is given by the following:(2)$$\gamma \left(t,\Sigma \left(t\right),\sigma_{y}\right)=\{\begin{array}{c} \;\;\;\;\;1,\;\;\;\;\;\;\;\;\;\;\;\;\;\;\;\;\;\;\;\;\;\;\;\;\;\;\;\;\;\;\;\;\;\;\;\;\;\;\;\;\;\;\;\varphi \left(t,\Sigma \left(t\right),\sigma_{y},\tau_{f},\alpha_{f}\right)\;\leq 1,\\ {\left(\varphi \left(t,\Sigma \left(t\right),\sigma_{y},\tau_{f},\alpha_{f}\right)\right)}^{-1/\alpha },\;\;\varphi \left(t,\Sigma \left(t\right),\sigma_{y},\tau_{f},\alpha_{f}\right)>1. \end{array}$$

Here, *t* is time, *Σ*(*t*) is the stress function of time, *σ_y_* is the static yield stress, and *α* is a dimensionless parameter of the material’s amplitude sensitivity. Inequality $\frac{1}{\tau }\int_{t-\tau }^{t}{\left(\frac{\Sigma \left(t_{1}\right)}{\sigma_{y}}\right)}^{\alpha }dt_{1}\;\leq 1$ is the incubation time criterion of yielding [34,42], equality in which assign the time of the beginning of plastic yielding for an arbitrary pulse. The equality *γ*(*t*) = 1 in Equation (2) corresponds to the case of accumulating elastic deformation before the onset of the macroscopic yield at *t*_*_. A decrease in the relaxation function in the range of 0 < *γ*(*t*) < 1 corresponds to the transition of the material to the plastic deformation stage. During the plastic deformation stage, where *t* ≥ *t*_*_, the following condition is satisfied for *γ*(*t*):(3)$$\gamma \left(t,\Sigma \left(t\right),\sigma_{y}\right)\varphi \left(t,\Sigma \left(t\right),\sigma_{y},\tau_{f},\alpha_{f}\right)=1$$

Equality in Equation (3) is retained because the state is fixed at the yield moment when *t* = *t_*_* (a detailed calculation scheme for *t*_*_ is given in [31]), and the accumulated elastic stresses are subsequently relaxed in the material (0 < *γ*(*t*) < 1). True stresses, *σ_true_*(*t*, *Σ*(*t*)), are determined by the following form:(4)$$\sigma_{true}\left(t,\Sigma \left(t\right)\right)=\psi \left(t,\Sigma \left(t\right),{Ε}_{mod},\sigma_{y},\beta \right)\varepsilon \left(t\right),\;\;\psi \left(\theta ,\Sigma \left(\theta \right),Β,\sigma_{y},a_{\gamma }\right)=Β{\left\{\gamma \left(\theta ,\Sigma \left(\theta \right),\sigma_{y}\right)\right\}}^{1-a_{\gamma }}\;$$
where *ψ*(*t*, *Σ*(*t*), *E_mod_*, *σ_y_*, *β*), is the coefficient related to the behaviour of stresses and *β* is the dimensionless scalar parameter (0 ≤ *β* < 1), which describes the degree of hardening of the material. Where *β* = 0, this corresponds to the absence of hardening. Considering the stages of elastic and plastic deformations separately, the general stress-strain relationship from Equation (4) is written as follows:(5)$$\sigma_{true}\left(t,\Sigma \left(t\right)\right)=\{\begin{array}{c} \;\;\;\;\;E_{mod}\varepsilon \left(t\right),\;\;\;\;\;\;\;\;\;\;\;\;\;\;\;\;\;t<t_{*},\\ E_{mod}\varepsilon \left(t\right)\gamma {\left(t,\Sigma \left(t\right),\sigma_{y}\right)}^{1-\beta },\;\;\;\;t\geq t_{*}. \end{array}$$

The set of fixed parameters used in drawing the deformation curve is independent of the strain rate and is only related to the changes in the material’s structure. The method for determining the parameters is considered in Selyutina and Petrov 2022 [31].

## 4. Plasticity Relaxation Model for Layered Materials

The multistage irreversible deformation of the multilayer composites under tension is studied based on the model outlined by Selyutina [27,28]. Utilizing the structural−temporal approach, Selyutina and Petrov [43] suggested an incubation time-based criterion of dynamic yielding that turned out to be an effective tool for predicting the limiting loading parameters of plastic deformation and to explain the instability of the strain rate dependencies on the dynamic yield point. One of the basic conclusions that were already driven by previous papers utilizing the incubation time criterion of yielding is that the strain-rate effect on any material yield strength could not be considered as an intrinsic material property. These investigations eventually resulted in the relaxation theory of plasticity [29,43]. Let us briefly consider the numerical scheme of the model:(6)$$\sigma_{true}\left(t\right)=\{\begin{array}{c} {Ε}_{0}\varepsilon \left(t\right),\;\;\;\;\;\;\;\;\;\;\;\;\;\;\;\;\;\varepsilon /\dot{\varepsilon }<t_{y}^{eff},\\ \psi \left(t,E_{0}\dot{\varepsilon }t,E_{0},\sigma_{y}^{eff},\beta_{0}\right)\varepsilon \left(t\right),\;\;\;\;\;\;t_{y}^{eff}\leq \varepsilon /\dot{\varepsilon }<t_{*}^{1},\\ \psi \left(t,E_{1}\dot{\varepsilon }t,E_{0},\sigma_{y}^{Me},\beta_{1}\right)\varepsilon \left(t\right),\;\;\;\;\;\;\;t_{*}^{1}\leq \varepsilon /\dot{\varepsilon }<t_{*}^{2},\\ \ldots \\ \psi \left(t,E_{n-1}\dot{\varepsilon }t,E_{0},\sigma_{y}^{Me},\beta_{n-1}\right)\varepsilon \left(t\right),\;\;\;\;\;\;\;\varepsilon /\dot{\varepsilon }>t_{*}^{n}. \end{array}$$

Here, *σ*(*t*) is the temporal dependence of the stress of the fibre metal laminate, *E_0_* is the initial Young’s modulus of the fibre metal laminate, *ε*(*t*) is the temporal dependence of the strain of fibre metal laminate, $\dot{\varepsilon }$ denotes the strain rate, *t_y_^eff^* is the yielding time of the metal layers, *γ*(*t*) is the relaxation function, *t_*_*^1^, *t_*_*^2^ … *t_*_^n^* are the fracture times of the brittle fibre layers of the composite, *E*_1_, *E*_2_, … *E_n−_*_1_ are the Young’s modulus of fibre metal laminate upon fracture of brittle 1… *n* − 1 fibrous layers, *n* is the number of fibrous layers, *β*_1_, *β*_2_, … *β**_n−_*_1_ are the degree of hardening of fibre metal laminate upon fracture of brittle 1… *n* − 1 fibrous layers, and *Σ*(*t*) is the temporal dependence of the load function in the specimen of the fibre metal laminate. Based on this function we built a model (Section 3) that predicts various forms of reactions of homogeneous materials to dynamic loading. The critical times, *ty^eff^* and *t_*_^i^*, were determined on the basis of the structural-time approach (1) written for the metal layers, as follows:(7)$$\varphi \left(t,\Sigma \left(t\right),\sigma_{y}^{eff},\tau_{y}^{eff},\alpha_{eff}\right)=1$$
and brittle layers:(8)$$\varphi \left(t,\Sigma \left(t\right),\sigma_{f}^{i},\tau_{f}^{i},\alpha_{f}^{i}\right)=1$$
where $\tau_{f}^{i}$ and $\tau_{y}^{eff}$ are the characteristic times of the fibre layers and metal layers, and *α_eff_* and $\alpha_{f}^{i}$ are sensitivity parameters of the material to the amplitude of the force field causing irreversible deformation of the aluminium and fibrous layers. Structural−temporal parameters α, α*^eff^*, $\alpha_{f}^{i}$, $\tau_{f}^{i}$, $\tau_{y}^{eff}$ and *β_i_* have constant values at any strain rate and different values for structurally different materials. $\sigma_{y}^{Al}$ and $\sigma_{f}^{i}$ are the static yield strength of the aluminium and the static strength of the brittle fibres, respectively. Detailed physical interpretations of the model parameters are given in following papers [27,30,34,43]. This model takes into account the change in the elastic modulus of the composite from *E_or_* to *E_fin_* upon fracture of the brittle fibre layers and the change in the cross-sectional area of FML from *S_or_* to *S_fin_*, as follows:(9)$$E_{or}S_{or}=E_{fin}S_{fin}$$

The process of deformation of a multilayer composite can be represented in the form of a rheological model, as schematically shown in Figure 1. In this paper, the St. Venant element (sliding element), designated as *p* element in Figure 1, is given by the homogeneous relaxation model of plasticity, just like in the paper of [31]. The model consists of a combined AA body (effective compound material), which describes the deformation of an aluminium alloy, and Hooke elastic bodies connected in parallel to it, which describe the deformation of fibres. The AA body consists of two Hooke elastic bodies and a plastic *p* body, and models elastic−plastic deformation with hardening. The yielding point $t_{y}^{eff}$, determined by Equation (7), defines the reference point of the deformation of the plastic *p* body in two stages, $\varepsilon \neq \dot{\varepsilon }\neq 0$, to which the deformation is zero $\varepsilon =\dot{\varepsilon }=0$. The general rheological equation of the model in Figure 1 is divided by the start point of the plastic flow, as follows:(10)$$\sigma_{eff}=\{\begin{array}{c} E_{eff}\varepsilon_{eff},\;\;\;\frac{1}{\tau_{eff}}\int_{t_{*}-\tau_{eff}}^{t_{*}}{\left(\frac{\Sigma \left(s\right)}{\sigma_{eff}^{y}}\right)}^{\alpha_{eff}}ds<1\\ E_{eff}\varepsilon_{eff}+\left(E_{eff}-E_{h}-E_{f}\right)\left(\varepsilon_{eff}-\varepsilon_{p}\right),\;\;\;\frac{1}{\tau_{eff}}\int_{t_{*}-\tau_{eff}}^{t_{*}}{\left(\frac{\Sigma \left(s\right)}{\sigma_{eff}^{y}}\right)}^{\alpha_{eff}}ds\geq 1 \end{array}$$

To establish the relationship between the parameters of the relaxation model of plasticity and the rheological model of the composite in Figure 1, there are two limiting cases of deformation:(11)$$E_{h}+E_{f}\overset{\beta =0}{\to }0$$
(12)$$E_{h}+E_{f}\overset{\beta =1}{\to }{Ε}_{eff}$$

Delamination between aluminium layers and fibres leads to a sharp drop in stress, corresponding to the removal of Hooke’s elastic elements, $E_{f}^{i}$. A stepped stress drop is observed with a multilayer composite with two or more prepreg layers.

Using the proposed model, it is possible to calculate the elastic, inelastic, and irreversible stages of composite deformation until the composite is completely destroyed. The proposed relaxation model for layered composites makes it possible to effectively describe the deformation response of the composite from elastic deformation to irreversible deformation, and to investigate further until the moment of failure. Moreover, this approach allows us to consider the effects of transition from static to dynamic loading, in which the dynamic limiting characteristics of the metal yield strength and the strength of brittle materials will have a determining character, depending on the loading history *Σ*(*t*). A united set of characteristic plasticity and fracture characteristic times also makes it possible to effectively predict important competitive phenomena in materials with a complex structural structure.

## 5. Results and Discussions

In this section, the structural−temporal parameters of the composite and its components are determined for metal composites with different structures. Table 1 lists the materials [17,18,19,20,44,45,46] used for numerical simulation of the plasticity relaxation model for layered materials. We used the notations Me/F/Me/F/Me or Me/F/Me for general definition of the sequence of metal (Me) and polymer fibre/epoxy layers (F). For instance, the abbreviation Me/F/Me/F/Me means that the fibre metal laminate consists of three metal layers and two polymer layers. We used the following abbreviations for notations materials: GFRP—glass fibre reinforced polymer; CFRP—carbon fibre reinforced polymer; KFRP—Kevlar fibre reinforced polymer; (0/90/90/0), (0/0/0/0), (0/90), and (90/0)—sequence of polymer layers with orientations of 0 and 90.

Figure 2, Figure 3 and Figure 4 show the theoretical deformation curves of different fibre metal laminates, plotted by experimental data in the studies [17,18,19,20,44,45,46]. The verification of the proposed relaxation model, as an example of the static and dynamic deformation dependencies, is carried out for various layered composites: Al/GFRP/Al/GFRP/Al, Ti/GFRP/Ti/GFRP/Ti, Al/GFRP/Al, Ti/GFRP/Ti, Al/CFRP/Al/CFRP/Al, and Al/KFRP/Al/KFRP/Al. Each of the presented static and dynamic theoretical dependencies shows the model performance for two composites: with different thickness ratios for the metal and polymer layers (Figure 2), with different metal layer materials for the composite (Figure 3), and with different polymer layer materials for the composite (Figure 4). The structural−temporal parameters of the layered composites for calculation of the theoretical dependencies are presented in Table 2, Table 3, Table 4, Table 5 and Table 6. The parameters of the relaxation plasticity model are invariant to the loading history, and using the table data, one can predict the rate sensitivity of the composites with an increase in the strain rate on the theoretical deformation dependencies (Figure 2, Figure 3 and Figure 4). Table 3 and Table 4 show that the same structural−temporal parameters for the Ti/GFRP(0/90/90/0)/Ti and Ti/GFRP (0/90)/Ti/GFRP(90/0)/Ti laminates and GFRP (0/90/90/0) of the polymer layers of composites, presented in Figure 2b and Figure 3, are used. The multistage of irreversible deformation of the layered composites are conditionally divided into three stages: elastic, nonlinear viscous-elastic, and preparation to fracture.

The pre-fracture stage may consist of the fracture of the polymer epoxy/glass fibre layers (including delamination process), which is different depending on the thickness of the composite layers (Figure 2). As the thickness of the metal layers in comparison with the polymer epoxy/fibre layers was greater in [44] (Figure 2a) than in [19,20] (Figure 2b), the pre-fracture stage on theoretical and experimental deformation curves had a stress drop at a strain rate of $\dot{\varepsilon }$ = 386 s^−1^ (Figure 2a), and continued to experience plastic deformation of the retained metal layers until fracture of the polymer epoxy/fibre layers at strain rates of $\dot{\varepsilon }$ = 0.001 s^−1^ and $\dot{\varepsilon }$ = 500 s^−1^ (Figure 2b).

The theoretical static and dynamic deformation curves, marked by solid lines in Figure 2, have a good correspondence with the static and dynamic experimental deformation curves, marked by symbols (Figure 2a,b), which verify that the relaxation model for the layered materials is capable of simulating two initial stages of deformation and both cases of pre-fracture stages. Note that to model of the theoretical deformation dependencies in Figure 2a and Figure 4, Equation (9) was not taken into account and the stress drops were also plotted as the thickness of the metal layers was greater than that the polymer epoxy/fibre layers.

The two-stage stress drop for statics (in Figure 2b) differs from the one-stage stress drop for dynamics. Continuing the dynamic theoretical dependence in Figure 2b, the second step of stress drop is also observed. The characteristic relaxation time of polymer glass fibre/epoxy layers (11 s in Table 2; 15,400 s and 100 s in Table 3) is usually several orders of magnitude higher than that of metal (1 ms in Table 2; 8.4 ms in Table 3); therefore, the fracture processes proceed faster. Figure 5 shows a comparison of the strain rate dependence of the strength for Ti-6Al-4V, calculated using the structural−temporal approach in [47], and the strain rate dependencies of the polymer fibre/epoxy strength in Table 3. Thus, the disappearance of the second stress release for the dynamic deformation curve of the composite with a high percentage of polymer can be explained by the predominance of preparatory processes for the fracture of these polymer layers in comparison with metal layers. Thus, the use of the technique for determining the characteristic times of the destruction of polymeric materials, the characteristic relaxation times of the metal material that is part of the composite, and the characteristic time of the composite make it possible to simultaneously take into account competing processes such as ductile−brittle transition, phase transition, and transient processes in the composite for different impact rates and materials of different structures.

Invariant structural−temporal characteristics of the loading history *Σ*(*t*) allow for the application of the proposed model for various combinations of metal and polymer layers. As shown in Table 3, theoretical deformation curves in Figure 3 are plotted with the same structural−temporal model parameters as the polymer glass/fibre layer [17,18,19,20]. The characteristic time of the Ti-based layered composite (8.4 ms in Table 3) is greater than the characteristic time of the Al-based layered composite (10 μs in Table 4). By changing only the structural−temporal parameters of the aluminium alloy to the titanium alloy, the theoretical deformation dependences of composites with one polymer layer, but a different metal matrix, are calculated on the basis of the proposed model with parameters (Table 2 and Table 3). The absence of smooth stress relief for the theoretical dynamic curve in Figure 3a is explained by taking into account the processes of destruction, without the process of delamination of the fibrous layers.

Typical dynamic dependences of layered composites with carbon [45] and Kevlar fibres [46] are also predicted based on the proposed model (Figure 4). Irreversible deformation of layered composites and the level of stress relief at the preparatory stage to fracture corresponded well with the experimental deformation curves [45,46]. A slight increase in the deformation curve at the second non-linear viscous-elastic stage of deformation of the layered composite was modelled using the structural−temporal parameters.

## 6. Conclusions

The modified plasticity relaxation model for heterogeneous materials was applied and verified for metal layered composites with different metallic and polymeric (fibre/epoxy) materials, tested under static ($\dot{\varepsilon }$ = 0.001 s^−1^) and dynamic loads ($\dot{\varepsilon }$ = 100–1300 s^−1^). The proposed relaxation model for the layered composites makes it possible to consider the effects of the transition from static to dynamic loading, in which the dynamic limiting characteristics of the metal yield strength and the strength of brittle materials will have a determining character, depending on the loading history.

Based on the proposed model, the pre-fracture stages of the composite with stress drops and continuous plastic deformation of the unbroken metal layers were predicted. Based on the same structural−temporal parameters for five- (Ti/GFRP (0/90)/Ti/GFRP(90/0)/Ti) and three-layer (Ti/GFRP(0/90/90/0)/Ti) glass fibre reinforced titanium composites and polymer layers, one-stage and two-stage stress drops during the irreversible deformation of the composite under static and dynamic loading were simulated. The one-stage stress drop on the dynamic curve of the composite and the two-stage stress drop on the static curve of the composite were compared according to the competition principle of the temporal preparatory processes of the layers fracture stress of the polymer and the metal.

The various deformation responses of metal composites in a wide range of strain rates, characterised by brittle, ductile, or ductile−brittle fractures, were predicted by the relaxation model of plasticity for heterogeneous materials. To model the important competitive phenomena on composite materials, a united set of characteristic times for plasticity and fracture was used.

Thus, the use of the plasticity relaxation model is not limited to homogeneous materials, but can be very effective for layered composites. Using examples of the composites with different thickness layers and calculated characteristic relaxation times for polymer (fibre/epoxy) and metal layers, the presence of elastic−plastic deformation after the fracture of the polymer (fibre/epoxy) layers of the composite is interpreted.

We believe that the basic ideas regarding the incubation time approach and the relaxation plasticity can serve as an effective tool in developing a new experimental standard and corresponding numerical schemes accounting for the unstable behaviour of deformation diagrams in the composite materials and their different components.

## Figures and Tables

**Figure 1 materials-15-04271-f001:**
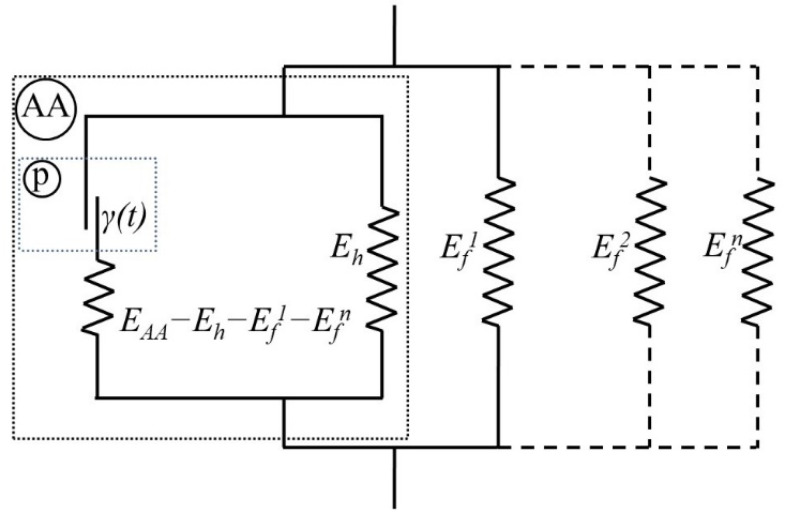
The rheological model of the fibre metal laminate.

**Figure 2 materials-15-04271-f002:**
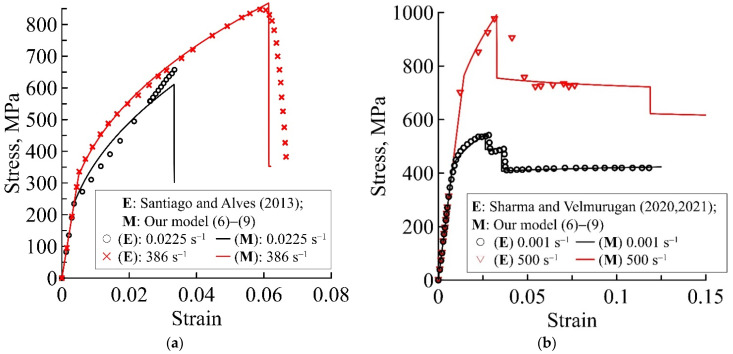
Predicted static and dynamic deformation curves of Me/GFRP/Me/GFRP/Me laminates using the proposed model (6)–(9), marked by solid lines, with different total thickness ratios of metal and polymer layers: (**a**) 1.2 and (**b**) 0.62. Experimental static and dynamic deformation curves for (**a**) Al/GFRP/Al/GFRP/Al [44] and (**b**) Ti/GFRP/Ti/GFRP/Ti [19,20] are marked by points.

**Figure 3 materials-15-04271-f003:**
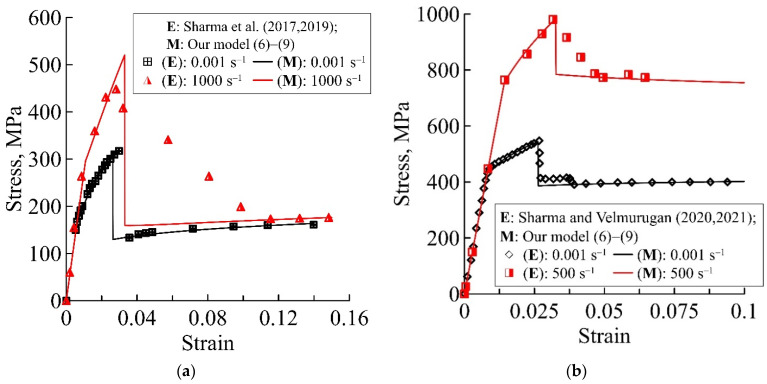
Predicted static and dynamic deformation curves of Me/GFRP/Me laminates using the proposed model (6)–(9), marked by solid lines, with close values for the total thickness ratio of metal and polymer layers: (**a**) 0.48 and (**b**) 0.63. Experimental static and dynamic deformation curves for (**a**) Al/GFRP/Al [17,18] and (**b**) Ti/GFRP/Ti [19,20] are marked by points.

**Figure 4 materials-15-04271-f004:**
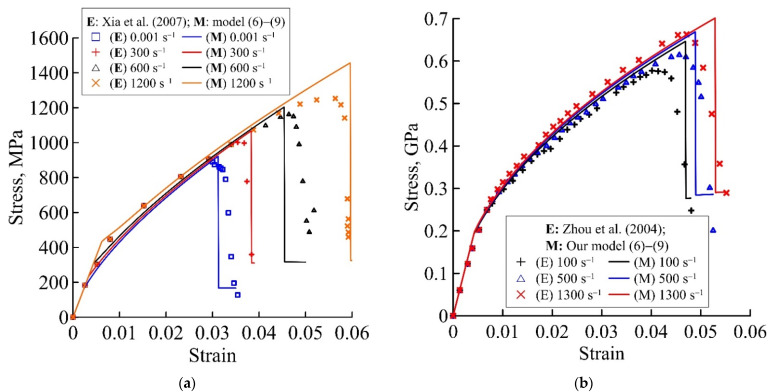
Predicted static and dynamic deformation curves of Al/CFRP/Al/CFRP/Al and Al/KFRP/Al/KFRP/Al laminates using the proposed model (6)–(9), marked by solid lines, with close values for the total thickness ratios for the metal and polymer layers: (**a**) 1.23 and (**b**) 1.16. Experimental static and dynamic deformation curves for (**a**) Al/CFRP/Al/CFRP/Al [45] and (**b**) Al/KFRP/Al/KFRP/Al [46] are marked by points.

**Figure 5 materials-15-04271-f005:**
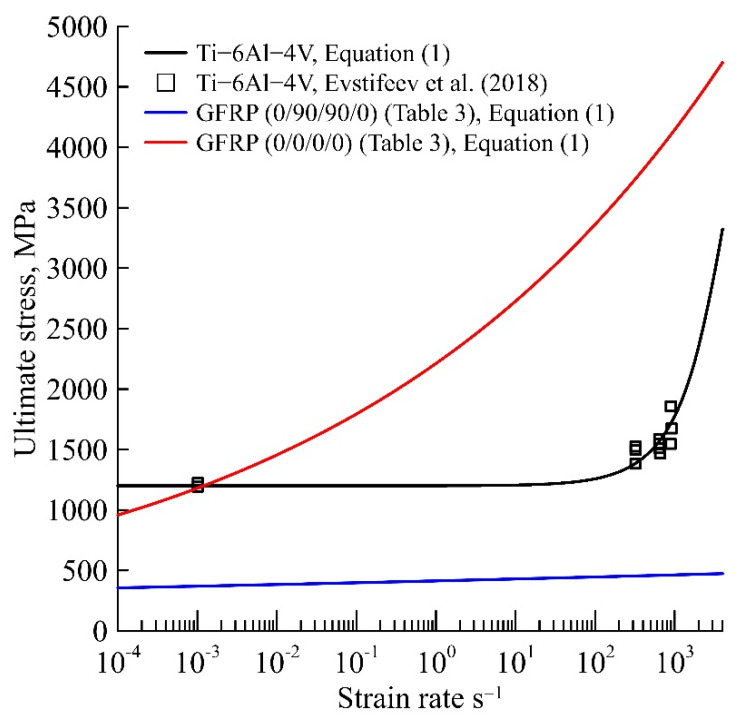
Comparison of strain rate dependencies of Ti-6Al-4V [47] and strain rate dependencies of the polymer fibre/epoxy (0/90/90/0) and (0/0/0/0) strength in Table 3.

**Table 1 materials-15-04271-t001:** Review of the layered materials utilised for numerical modelling using the relaxation model of plasticity for layered materials.

Reference	Composite Structure	Metallic Layer	Type of Fibre	Thickness One Me Layer, mm	Thickness of Composite, mm	Thickness Ratio of Metal and Polymer Layers
[44]	Me/F/Me/F/Me	Al 2024-T3 alloy	GFRP	0.4	2.2	1.2
[17,18]	Me/F/Me	Al 2024-T3 alloy	GFRP	0.6	3.7	0.48
[19,20]	Me/F/Me	Ti-6Al-4V alloy	GFRP	0.6	3.1	0.63
	Me/F/Me/F/Me		GFRP	0.4	3.15	0.62
[45]	Me/F/Me/F/Me	LY12CZ Al alloy	CFRP	0.246	1.34	1.23
[46]	Me/F/Me/F/Me	Al alloy	KFRP	0.24	1.34	1.16

**Table 2 materials-15-04271-t002:** Parameters of the model and materials [44] for the calculated deformation curves of the GLARE Al/GFRP/Al/GFRP/Al composite in Figure 2a.

Material	Property	Unit	Value
Al/GFRP/Al/GFRP/Al	$\sigma_{y}^{eff}$	MPa	235
	${Ε}_{0}$	GPa	65
	$\alpha_{eff}$	–	15
	$\tau_{y}^{eff}$	ms	1
	*β*	–	0.43
Al 2024-T3 alloy	$\sigma_{y}^{Me}$	MPa	302.29
	*β*	–	0
GFRP (0/90/90/0)	$\sigma_{f}^{1}$	MPa	340
	${Ε}_{f}$	GPa	14
	$\alpha_{f}^{1}$	–	1
	$\tau_{f}^{1}$	s	11

**Table 3 materials-15-04271-t003:** Parameters of the model and materials [17,18] for the calculated deformation curves of the GLARE Ti/GFRP(0/90/90/0)/Ti composite laminate and Ti/GFRP(0/90)/Ti/GFRP(90/0)/Ti laminate in Figure 2b and Figure 3b.

Material	Property	Unit	Value
Ti/GFRP(0/90/90/0)/Ti laminate or Ti/GFRP (0/90)/Ti/GFRP(90/0)/Ti laminate	$\sigma_{y}^{eff}$	MPa	430
	${Ε}_{0}$	GPa	53
	$\alpha_{eff}$	–	15
	$\tau_{y}^{eff}$	ms	8.4
	*β* at 0.001 s^−1^	–	0.2
	*β* at 500 s^−1^	–	0.36
Ti-6Al-4V alloy	$\sigma_{y}^{Me}$	MPa	984
	*β*	–	0.03
GFRP (0/90/90/0)	$\sigma_{f}^{1}$	MPa	310
	${Ε}_{f}$	GPa	14
	$\alpha_{f}^{1}$	–	60
	$\tau_{f}^{1}$	s	15,400
GFRP (0/0/0/0)	$\sigma_{f}^{2}$	MPa	665
	${Ε}_{f}$	GPa	33.26
	$\alpha_{f}^{2}$	–	10
	$\tau_{f}^{2}$	s	1000

**Table 4 materials-15-04271-t004:** Parameters of the model and materials [19,20] for the calculated deformation curves of the GLARE Al/GFRP/Al/GFRP/Al composite in Figure 3a.

Material	Property	Unit	Value
Al/GFRP/Al/GFRP/Al laminate	$\sigma_{y}^{eff}$	MPa	160
	${Ε}_{0}$	GPa	27.22
	$\alpha_{eff}$	–	1
	$\tau_{y}^{eff}$	μs	10
	*β* at 0.001 s^−1^	–	0.45
	*β* at 1000 s^−1^	–	0.65
Al 2024-T3 alloy	$\sigma_{y}^{Me}$	MPa	302.29
	*β*	–	0.14
GFRP (0/90/90/0)	$\sigma_{f}^{1}$	MPa	310
	${Ε}_{f}$	GPa	14
	$\alpha_{f}^{1}$	–	60
	$\tau_{f}^{1}$	s	15,400

**Table 5 materials-15-04271-t005:** Parameters of the model and materials [45] for the calculated deformation curves of the GLARE Al/CFRP/Al/CFRP/Al composite laminate in Figure 4a.

Material	Property	Unit	Value
Al/CFRP/Al/CFRP/Al laminate	$\sigma_{y}^{eff}$	MPa	160
	${Ε}_{0}$	GPa	70
	$\alpha_{eff}$	–	1
	$\tau_{y}^{eff}$	μs	7
	*β*	–	0.67
LY12CZ aluminium alloy	$\sigma_{y}^{Me}$	MPa	302.29
	*β*	–	0
CFRP (0/90/90/0)	$\sigma_{f}^{1}$	GPa	1
	${Ε}_{f}$	GPa	32
	$\alpha_{f}^{1}$	–	1
	$\tau_{f}^{1}$	μs	47.3

**Table 6 materials-15-04271-t006:** Parameters of the model and materials [46] for the calculated deformation curves of the Al/KFRP/Al/KFRP/Al composite in Figure 4b.

Material	Property	Unit	Value
Al/KFRP/Al/KFRP/Al laminate	$\sigma_{y}^{eff}$	MPa	170
	${Ε}_{0}$	GPa	46
	$\alpha_{eff}$	–	60
	$\tau_{y}^{eff}$	ms	1
	*β*	–	0.51
Aluminium alloy	$\sigma_{y}^{Me}$	MPa	302.29
	*β*	–	0
ΚFRP (0/90/90/0)	$\sigma_{f}^{1}$	GPa	1.3
	${Ε}_{f}$	GPa	28
	$\alpha_{f}^{1}$	–	1
	$\tau_{f}^{1}$	μs	10

## Data Availability

Not applicable.

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
