# Peer review of "Structural-Temporal Peculiarities of Dynamic Deformation of Layered Materials"

_materials, 2022, doi:10.3390/ma15124271_

Round 1

Reviewer 1 Report

Authors studied the structural-temporal peculiarities of dynamic deformation of 2 layered materials. This study is very interesting and very useful to scientific community. Hence this paper is recommended for publication after minor revision.

  • Me/F/Me/F/Me – define abbreviations first time in the text before introducing the text or table.
  • Authors used figure 3 and 4 form other reference works. How these graphs are used when the tests had been conducted in different machines and different parameters?
  • Table 2 - 6 – provide units clearly in separate column
  • The strength values are provided. But have you considered the experimental error?
  • Check the language of the paper.
  • Experimental validation of the model is missing.
  • Mention the novelty of the work at the end of the introduction section.
  • Cite few more recent references the revised paper.
  • Numerical values are absent in the conclusion as well as in the abstract section.
  • Figure 5- change the line time type or marker type
  • Improve the discussion with respect to the change in strain rate

Author Response

Dear Rev.1,

Thank you for opportunity to present a revised version of our manuscript for reconsideration. We are thankful to Rev. 1 for useful comments that allow us to improve our presentation; bellow all of them are cited and answered. All changes are marked out by red in the revised manuscript.

Truly yours,

Authors

See attached file

Reviewer 2 Report

The paper "Structural-temporal peculiarities of dynamic deformation of layered materials" presents a topic adherent to this journal, however in its current stage it is not suitable for publication, comments that justify this decision are described below:

(1) The topic in general is not presented by the authors with any innovation and/or significant contribution to the state of the art, this harms the paper and its potential for future citation, which certainly impacts on quality indicators;
(2) The authors cannot make a consistent and adequate review of the literature, proof of this is that most of the existing references are old, this is harmful to the paper and certainly diminishes its amplitude in the international scientific community;
(3) The abstarct does not bring relevant methodological aspects and main quantitative results, immediately readers are discouraged from reading the paper!!
(4) The results are often presented as a sequence, without in-depth discussion with new papers and good journals!

For all these reasons I must recommend the rejection of this paper at its current stage, encouraging improvements by the authors.

Author Response

Dear Rev.2,

Thank you for opportunity to present a revised version of our manuscript for reconsideration. We are thankful to Rev. 1 for useful comments that allow us to improve our presentation; bellow all of them are cited and answered. All changes are marked out by red in the revised manuscript.

Truly yours,

Authors

See attached file

Reviewer 3 Report

see attachment.

I look forward to your next version.

Author Response

Dear Rev.3,

Thank you for opportunity to present a revised version of our manuscript for reconsideration. We are thankful to Rev. 1 for useful comments that allow us to improve our presentation; bellow all of them are cited and answered. All changes are marked out by red in the revised manuscript.

Truly yours,

Authors

See attached file.

Round 2

Reviewer 2 Report

The paper "Structural-temporal peculiarities of dynamic deformation of layered materials" has been improved by the authors, but there are still corrections:

(1) The literature was reinforced by the authors, however some studies can still be inserted as: 10.4028/www.scientific.net/MSF.798-799.548; 10.1016/j.cscm.2022.e01129; 10.1016/j.cscm.2022.e01182.

(2) Some legends as in Figure 2 are extensive, it is not common to have references in legends! This is repeated in several other figures;

(3) The sequence of tables is not good, authors should introduce explanatory texts and discussions between them;

(4) I understand that the conclusion has improved, often at the request of the other reviewers, but it is too long for a conclusion, this should be strongly checked.

Author Response

(1) The literature was reinforced by the authors, however some studies can still be inserted as: 10.4028/www.scientific.net/MSF.798-799.548; 10.1016/j.cscm.2022.e01129; 10.1016/j.cscm.2022.e01182.

The authors are not sure that these articles on concrete can be on the topic of the article. These papers do not even consider dynamic loads, multilayer composites or any models. But if the reviewer cannot accept the work without references to these works, then we have added them.

(2) Some legends as in Figure 2 are extensive, it is not common to have references in legends! This is repeated in several other figures;

All legends of Fig. 2-Fig.5 references to experiments were put down by numbers [44] instead of a record (Santiago and Alves, 2013).

(3) The sequence of tables is not good, authors should introduce explanatory texts and discussions between them;

All Tables in the manuscript are given in the correct sequence. Sufficient discussion of tables is already present in the paper. The tables in the work show the obtained structural-temporal parameters for obtaining deformation dependences according to the proposed model, which are constant in Fig. 2, Fig.3, Fig. 4. This is their main role. Listing the obtained structural-temporal parameters in the text without a table is worse than with a table. The corresponding in the caption of each Table is a link to the Figure, the deformation dependences of which are built according to the proposed model.

In this paper, the proposed model is verified on the basis of various composites. Each of the presented figures demonstrates how the same model predicts different static and dynamic deformation dependences: (Fig. 2) for two composites with different thicknesses of the polymer and metal layers of the composite, (Fig. 3) for two composites with different materials of the metal layers, (Fig. 4) for two composites with different polymer fiber materials. Figure 2 considers composites with different ratios of the thickness of metal layers and polymer layers, which leads to the presence or absence of the stage of plastic deformation of the metal layers after the destruction of the polymer layers of the composite. Figure 3 considers the case of an ongoing flow stage using composites as an example. Figure 4 considers the case of failure of the composite immediately after the failure of the polymer layers of the composite. The difference in model verification in Fig.3 and Fig. 4 in that the materials have different fixed parameters determined by a single algorithm according to the model, but different deformation dependences with the destruction of the composite immediately after the destruction of the polymer layers (Figure 4) or after the ongoing stage of plastic deformation of metals (Figure 3) are obtained automatically. Since the discussion of the results is carried out simultaneously for various composites with completely different deformation behaviors according to the Figures, then requiring the authors to discuss each table separately is a violation of the overall built-in logic of the work and a deviation from the general idea of the work: by determining the structural-temporal parameters for the composite and its components, we can predict different deformation responses.

(4) I understand that the conclusion has improved, often at the request of the other reviewers, but it is too long for a conclusion, this should be strongly checked.

We shortened the conclusion a little without losing the general meaning.

Reviewer 3 Report

It shoule be accepted.

Author Response

The English language of the manuscript has been significantly improved.